# HEHR: Homing Endonuclease-Mediated Homologous Recombination for Efficient Adenovirus Genome Engineering

**DOI:** 10.3390/genes13112129

**Published:** 2022-11-16

**Authors:** Katrin Schröer, Fatima Arakrak, Annika Bremke, Anja Ehrhardt, Wenli Zhang

**Affiliations:** Virology and Microbiology, Center for Biomedical Education and Research (ZBAF), Department of Human Medicine, Faculty of Health, Witten/Herdecke University, 58448 Witten, Germany

**Keywords:** adenovirus vector, adenoviral genome, seamless mutagenesis, homologous recombination, I-PpoI endonuclease

## Abstract

Adenoviruses are non-enveloped linear double-stranded DNA viruses with over 100 types in humans. Adenovirus vectors have gained tremendous attention as gene delivery vehicles, as vaccine vectors and as oncolytic viruses. Although various methods have been used to generate adenoviral vectors, the vector-producing process remains technically challenging regarding efficacious genome modification. Based on our previously reported adenoviral genome modification streamline via linear–circular homologous recombination, we further develop an HEHR (combining Homing Endonucleases and Homologous Recombination) method to engineer adenoviral genomes more efficiently. I-PpoI, a rare endonuclease encoded by a group I intron, was introduced into the previously described *ccdB* counter-selection marker. We found that the I-PpoI pre-treatment of counter-selection containing parental plasmid increased the homologous recombination efficiency up to 100%. The flanking of the counter-selection marker with either single or double I-PpoI sites showed enhanced efficacy. In addition, we constructed a third counter-selection marker flanked by an alternative restriction enzyme: AbsI, which could be applied in case the I-PpoI site already existed in the transgene cassette that was previously inserted in the adenovirus genome. Together, HEHR can be applied for seamless sequence replacements, deletions and insertions. The advantages of HEHR in seamless mutagenesis will facilitate rational design of adenoviral vectors for diverse purposes.

## 1. Introduction

The Adenovirus vector is one of the most studied vectors in the research laboratory and is among the most widely used reagents in gene therapy clinical trials. As stated in the Journal of Gene Medicine, recombinant adenoviruses are accounting for 17.5% of vectors used in gene therapy clinical trials [1]. Adenovirus vectors have been broadly explored as oncolytic viruses to treat malignant tumors, as a genetic vaccine against infectious diseases and as a gene transfer vectors for gene therapy applications. Several features make the adenovirus an advantageous viral vector, such as the stable genetic material (double-stranded linear DNA) and outstanding replicating ability with its own high proofreading DNA polymerase. Moreover, the non-enveloped virus, adenovirus, has a proliferation cycle concluding in destruction of cells in the host organism. This destructive proliferation activity is an advantage for oncolysis compared to envelope viruses, which often complete proliferation by growing from vital, undamaged host cells. Another advantage of the adenovirus is the fact that it comprises plentiful types in humans, making it flexible and versatile toolbox for optimized vector design. To date, 113 human adenovirus types (Ad1 to Ad113) have been identified and classified into seven species (A to G) based on hemagglutination properties, oncogenicity in rodents, DNA homology and genome organization [2].

To convert adenoviruses to vectors, several strategies have been devised by different research groups. So far, over one-third of the 113 identified human adenoviruses have been converted to vectors [3,4,5]. After the initial vectorization, further genetic modification is needed for deeper studies of these adenoviruses, as well as vector tool development. Although various methods have been used to generate adenovirus vectors, the adenovirus-producing process remains technically challenging regarding efficacious genome modification. We have previously developed high-throughput adenoviral genome direct cloning via linear–linear homologous recombination (LLHR) and genome modification via linear–circular homologous recombination (LCHR) (Figure 1) [6,7]. With this strategy, various new adenovirus vectors have been generated for receptor studies, as oncolytic viruses for tumor treatment, as gene delivery vector for gene therapy, as well as novel immunotherapeutic strategies to combat severe adenovirus infection in immunocompromised hosts [8,9,10,11,12].

However, some modifications associated with high GC-content or large inserts showed to be more challenging. The work described here was encouraged by the fact that a double-strand break generated by CRISPR/Cas9 can highly increase the homologous recombination efficiencies in mammalian genomes [14]. Therefore, we hypothesized that homologous recombination efficiencies in bacteria (*E. coli*) can also be improved by introducing a double-strand break with a restriction enzyme. We pre-scanned most of identified human adenovirus genomes, and filtered out the non-cutters from each species (Table 1 and Appendix A). Observing only homing endonucleases were shared among all different species, we chose I-PpoI, a rare endonuclease encoded by a group I intron [15], for further application. We chose this I-PpoI based on two simple reasons: This endonuclease has the shortest recognition sequence (15 bp) among the five available enzymes, which makes the selection marker plasmid construction convenient. Moreover, the other four endonucleases were quite often used as enzymes to release the adenoviral genome for virus rescue, such as I-CeuI and I-SceI already positioned in our p15A-based vector backbone [7].

In the current protocol, we describe a HEHR method by combining Homing Endonucleases and Homologous Recombination to engineer adenovirus genomes more efficiently. We first introduced I-PpoI recognition site into the previously described *ccdB* counter-selection marker (SM) plasmid by flanking the SM either on both or one site with the I-PpoI recognition site [7,16] (Appendix A). We found that the I-PpoI pre-treatment of the counter-selection marker containing parental plasmid increased the homologous recombination efficiency up to 100%. The flanking of counter-selection marker with single or double I-PpoI sites revealed no difference regarding cloning efficiency. HEHR can be applied for seamless sequence replacements, deletions and insertions (Figure 2). The advantages of HEHR in seamless mutagenesis will facilitate the rational design of adenovirus vectors for diverse purposes from basic virology study to application fields, including gene therapy, vaccine trials and cancer treatments as oncolytic viruses.

## 2. Materials

### 2.1. Reagents

All essential reagents used in this protocol are listed in Table 2.

### 2.2. Essential Equipment

The protocol presented here does not require any special equipment; the work can be performed in a normal molecular laboratory. Below is the list of equipment used: 

Bacteria incubator (37 °C); Bench-top centrifuge with cooling function; Biological safety cabinet; Electroporator; Micro-centrifuge tube, 1.5 mL and 2 mL; Micro-tube thermal mixer; Falcon Tube, 15 mL and 50 mL; Freezer (−20 °C); Gel electrophoresis and imager system; pipettes and tips; Refrigerator (4 °C); Spectrophotometer; Syringe needle (25 G) and Vortex mixer

## 3. Results with Detailed Protocol

Since the adenovirus genome engineering with the original selection marker (SM) was already described previously [7,16], in the current protocol we focused on the HEHR method (combining Homing Endonucleases and Homologous Recombination), and share the detailed protocol.

A schematic outline for adenovirus genome engineering with HEHR is illustrated in Figure 3. The I-PpoI flanked SM is first inserted into the target adenoviral genome in the first week. After the I-PpoI mediated linearization of the SM-containing adenoviral plasmid, the insert of choice replaces the SM in the second week, resulting in the intended construct. Afterwards, the recombinant adenoviral vector can be reconstituted by transfection of the adenovirus genome released from the plasmid and subsequent virus amplification in around 3 weeks, which is then ready for the in vitro/vivo application.

### 3.1. Insert the Counter-Selection Marker (SM) ccdB-Amp

#### 3.1.1. Oligonucleotide Design for SM Flanked by Homology Arms

In principle, for the forward primer, choose 50 nucleotides directly adjacent upstream (5′) to the intended insertion site. Order an oligonucleotide with this sequence at the 5′ end. At the 3′ end of this oligonucleotide, the PCR primer sequence to amplify the *ccdB-Amp* cassette should be included. For the reversed primer, choose 50 nucleotides directly adjacent downstream (3′) to the intended insertion site and transfer them into the reverse complement orientation. Order an oligonucleotide with this sequence at the 5′ end. At the 3′ end of this oligo, include the 3′ PCR primer sequence (also in reverse complement orientation) for the *ccdB-Amp* cassette.

Optional: use proper DNA analysis software to perform in silico cloning to generate the final construct. (1) Copy the ~50 nucleotides from the target genome sequence in upstream orientation to the target site as a homology arm that is continued with primer sequence for amplification of the *ccdB-Amp* cassette. (2) Copy the reverse complement of the ~50 nucleotides from the adenoviral genome sequence downstream of the target site as homology arm that is continued with primer sequence for amplification of the *ccdB-Amp* cassette. Figure 4A shows an example of a primer design for inserting GFP into the E1 region of adenovirus type 5 (Ad5).

We then perform PCR from the plasmid pR6K-(IP)-ccdB-Amp-(IP), which offers an ampicillin-resistant gene as a positive selection marker and *ccdB* as a counter-selection marker. This SM is flanked by I-PpoI site (Appendix A). The R6K plasmid backbone avoids the cloning background from the parental plasmid.

#### 3.1.2. PCR Product Preparation

##### PCR Condition: With Primers SM-fwd and -rev from the pR6K-SM Plasmid

Perform PCR reaction according to the manufactory’s protocol. Below is the example using PrimeSTAR DNA polymerase (Takara) (Table 3 and Table 4).

Aliquot to two tubes (50 µL each), run PCR with the following program.

##### Purification of PCR Product

(1)Keep 5 µL for agarose gel check (a), purify the remaining PCR product according to the manufactory’s protocol.(2)After purification, keep 2 µL for agarose gel check (b), desalt the remaining product with nitrocellulose filters against dH2O for 30 min at room temperature.(3)Check the concentration and take 3 µL for agarose gel check (c).(4)Agarose gel checks the three collected aliquots (a), (b), (c)(5)After agarose gel confirmation and concentration measurement, aliquot the final product to 1000~2000 ng per tube, keep one tube at 4 °C for use up to two weeks, store the others at −20 °C.

#### 3.1.3. Linear-Circular Homologous Recombination (LCHR)

##### Day 1: Preparation of the Overnight *E. coli* Culture

One day before the direct cloning experiment, puncture the caps of three 1.5 mL tubes for aeration by a syringe needle and add 1 mL LB medium containing streptomycin (100 µg/mL). Inoculate a single colony from the *E. coli* GBRed-GyrA462 containing plate to each tube and incubate at 37 °C overnight with shaking at 1050 r.p.m. in a micro-tube thermal mixer.

##### Day 2: LCHR

(1) Inoculate 30–50 µL of the ON-culture (with OD600 between 3 and 4) into two new tubes with punctured caps, each containing 1.4 mL LB medium containing streptomycin (100 µg/mL). Incubate at 37 °C with shaking at 1050 r.p.m for about 1.5 h to achieve an OD600 between 0.35 and 0.40.

(2) Add 50 µL of 10% L-arabinose into one tube to induce recombinase protein expression, while in the second tube omit the addition of L-arabinose to generate a negative control. Continue the culture at 37 °C for 45 min. At this stage, the OD600 value should be between 0.7 and 0.8.

(3) Spin down the cells by centrifugation at 11,000 r.p.m. for 30 s, at 2 °C.

(4) Discard the supernatant by decanting and thoroughly resuspend the cell pellet in 1 mL of ice-cold sterile water by vortexing at the maximum speed.

CRITICAL STEP: Discard as much as possible from the supernatant, as residual salt will reduce the efficiency of electroporation. Take care to avoid losing the cell pellet.

(5) Repeat steps (3)-(4)-(3). In the end, only discard the supernatant, do not add more water.

(6) Add 500~1000 ng PCR generated SM (flanked by homology arms) and ~100 ng to-be-modified adenoviral genome containing plasmid into the tube, gently mix by pipetting up and down. Then, transfer the entire mixture into a pre-cooled 1 mm electroporation cuvette and keep it on ice.

(7) Electroporate at 1350 V, 10 µF, 600 Ω. This setting applies to an Eppendorf Electroporator Eporator using an electroporation cuvette with a 1 mm gap.

(8) Recovery for 1.5 h in 1 mL LB medium without antibiotic at 37 °C with shaking at 1050 r.p.m. in a micro-tube thermal mixer.

(9) Streak 100 µL of the culture onto a LB plate with both chloramphenicol (15 µg/mL) and ampicillin (50 µg/mL). Incubate at 37 °C overnight.

##### Day 3: Set Up Cell Cultures for Clone Identification

Set up six 2 mL tubes, puncture the cap for aeration, and add 1.8 mL LB medium with both chloramphenicol (10 µg/mL) and ampicillin (30 µg/mL). Inoculate each tube with a single colony, and culture at 37 °C overnight shaking with 1050 r.p.m. in a micro-tube thermal mixer.

##### Day 4–5: Cloning Verification by Restriction Enzyme Analysis

Plasmid isolation according to the manufactory’s protocol. Choose the restriction enzyme, from which the digest pattern can be differentiated from the parental plasmid. Re-transform the identified plasmid into *E. coli* GBRed-GyrA462 competent cells, and perform new mini-culture to isolate the plasmid (keep backup), digest to confirm. Figure 5 are some examples of restriction enzyme digest.

Figure 5A is a restriction enzyme digest example of SM insertion to delete E1 region and result in the plasmid pAd5-SMdE1. Figure 5B is an example of plasmid restriction digest of the re-transformed plasmids. Figure 5C is an example of another plasmid (pAdX) restriction digest (Data regarding unpublished adenovirus construction). In this example, clone number 6 is an incorrect plasmid. Figure 5D is the digest after re-transformation. The 6 kb band that is presented in Figure 5C disappeared after re-transformation. Figure 5C is a typical example of the co-existing of both parental and recombinant plasmids in the same clone.

Note: it is essential to re-transform the identified plasmids to get rid of the parental plasmid.

Optimal: one can perform sequence analysis to confirm the cloning junctions.

##### Day 6–7: (Optimal) Large-Scale Plasmid Preparation

Setup midi-culture from two backups (of colonies identified after re-transformation) with LB chloramphenicol (10 µg/mL) and ampicillin (30 µg/mL), perform midi-preps and OD measurement for concentration check, aliquot to ~5 µg/tube, keep one of each at 4 °C for use; store the others at −20 °C.

Optimal: one can perform sequence analysis to confirm the cloning junctions.

### 3.2. Replace the Counter-Selection Marker ccdB-Amp with the Aimed Gene

#### 3.2.1. Linearize the Plasmid (pAd-SM)

(1) Digest the pAd-SM with I-PpoI at 37 °C overnight (Table 5).

(2) On the next day, precipitate the DNA by adding 20 µL of 3 M sodium acetate (pH 5), 2 µg glycogen and 500 µL of precooled ethanol (≥99.8%; stored at −20 °C). Mix gently by inverting the tube several times.

Important: to increase the recovery rate, incubate the mixture at −20 °C for 30 min; never vortex the large linear DNA.

(3) Centrifuge for 20 min at full speed (≥15,000 g) in a micro-centrifuge and discard the supernatant.

(4) Add 600 µL of 70% ethanol and mix gently by inverting the tube several times. After centrifugation at full speed (≥15,000 g) incubate at room temperature for 10 min, remove the supernatant.

(5) Repeat step (4).

(6) Air-dry pellet briefly and suspend in 30 µL of sterilized water. Dissolve the DNA with low-speed shaking (300 r.p.m.) at room temperature for 15 min.

(7) Desalt the product with nitrocellulose filters for 30 min.

(8) Check the concentration.

(9) Aliquot to ~1000 ng/tube, keep one at 4 °C for use; store others at −20 °C.

#### 3.2.2. Preparation of PCR Product for Linear-Linear Homologous Recombination (LLHR)

Similar to Section 3.1.2, Figure 4B is an example of a primer design for inserting GFP into Ad5 E1 region.

#### 3.2.3. LLHR

##### Day 1: Preparation of the Overnight *E. coli* Culture

One day before the direct cloning experiment, puncture the caps of 3× 1.5 mL tube for aeration by syringe needle and add 1 mL LB medium containing streptomycin (100 µg/mL). Inoculate a single colony from the *E. coli* strain GB05-dir plate to each tube, and incubate at 37 °C overnight with shaking at 1050 r.p.m. in a micro-tube thermal mixer.

##### Day 2: LLHR

(1) Inoculate 35 µL of the ON-culture into a new tube with punctured caps containing 1.4 mL LB medium with streptomycin (100 µg/mL). Culture at 37 °C for ~1.5 h (1050 r.p.m.).

(2) Add 50 µL of 10% L-arabinose to induce the recombinase protein expression and then grow for another 45 min. In a second tube omit the addition of L-arabinose to generate a negative control.

(3) Make cells electrocompetent according to the same protocol used in Section 3.1.3.

(4) Electroporate the mixture of 1000 ng PCR product and 500 ng linearize DNA to each tube.

(5) Recovery for 1.5 h in 1 mL LB medium without antibiotic at 37 °C with shaking at 1050 r.p.m. in a micro-tube thermal mixer.

(6) Spin down (9000 r.p.m., 1 min) the recovered culture, discard the supernatant and re-suspend the pellet with 100 µL LB to plate 50 µL onto LB plate with chloramphenicol (15 µg/mL). Plate the bacteria in the dilution way. Incubate at 37 °C overnight.

##### Day 3: Set Up Cell Cultures for Clone Identification

Inoculate twelve single colony from each plate into 2 mL reaction tubes containing 1.8 mL LB with chloramphenicol (10 µg/mL). Inoculate a single colony of the parental plasmid as control. Incubate for overnight at 37 °C with 1050 r.p.m. shaking on an Eppendorf Thermomixer.

##### Day 4–5: Cloning Verification by Restriction Analysis

Plasmid isolation according to the manufactory’s protocol. Choose the restriction enzyme, from which the digest pattern can differentiate from the parental plasmid. Figure 6A shows a restriction digest example of GFP insertion to E1 region and results in the plasmid pAd5-dE1-GFP. Clones number 9 and 11 are positive, confirmed by all expected bands. Re-transform the identified plasmid into normal competent cells, and perform a new mini-culture to isolate the plasmid, digest to confirm. Figure 6B displays an example of plasmid restriction digest of the re-transformed plasmids. Here, nine from ten checked plasmids were correct, while clone number 1 contains intra-molecular-recombination. Figure 6C,D show examples of another plasmids (pAdX, pAdY) restriction digest (Data regarding unpublished adenovirus constructions).

Note: it is essential to re-transform the identified plasmids to get rid of the parental plasmid.

##### Day 6–7: Large-Scale Plasmid Preparation

Setup midi-culture from the re-transformed plasmid with LB chloramphenicol (10 µg/mL), perform midi-preps and OD measurement for concentration check, aliquot to ~10 µg/tube, keep one at 4 °C for use; store the others at −20 °C.

Optimal: one should perform the sequence analysis to confirm the cloning junctions and respective transgene.

After confirmation, the newly generated adenoviral genome containing plasmid can be used for virus production or further modification.

## 4. Discussion

We developed a HEHR system based on homing endonucleases and homologous recombination to facilitate the efficient construction of recombinant adenoviruses. In our novel system, the counter-selection marker (SM) containing adenovirus plasmid can be linearized by homing endonucleases, and then the aimed transgene expression cassette is incorporated into the desired position seamlessly via linear–linear homologous recombination (LLHR). Compared with conventional construction methods, the HEHR system increases the cloning efficiency, therefore reducing the time and labor, which allows for high-throughput recombinant adenovirus generation.

There is increasing interest in human adenoviruses from two distinguish directions: first of all is the persistent adenovirus infections in immunocompromised patients and outbreaks in children [20,21,22,23] and the second is due to the promising applications as efficient vectors for vaccine and gene therapy clinical trials [1,24,25,26,27,28]. For both purposes, it is essential to vectorize the adenovirus genome and construct recombinant novel vectors. Previously, we and other groups have applied different methods or platforms to vectorize and engineer individual adenovirus types, such as cosmid-based methods, traditional cut- and-paste based molecular cloning, homologous recombination in bacteria or in eukaryotic cells and most recently described Gibson gene assembly technique [5,6,7,29,30,31,32,33,34,35]. Each method has its advantage and disadvantage, in general, most methods were either time-consuming or complicated. To speed up the process from new type identification to vectorizations and recombinant virus production for respective applications, it is important to have a method with workable solution.

The optimized protocol we presented here was the accumulation of method development from the past decade. With the motivation to have a reliable cloning efficiency, inspired from CRISPR/Cas9 guided homologous recombination in mammalian genomes [14,36], we tried to introduce random non-cutters next to the selection marker. Thereby converting the ccdB-mediate counter selection from linear–circular homologous recombination (LCHR) to linear–linear homologous recombination (LLHR), which made some previous difficult/impossible cloning with large insertion or high GC content possible. To broaden the usage of LLHR for counter selection, we try to generate a universal counter-selection marker for all known human adenoviruses genome engineering. To identify such restriction enzymes, we scanned most of the identified human adenovirus genomes and found that only homing endonucleases were shared among all different species (Table 1 and Appendix A). Therefore, I-PpoI, a rare endonuclease encoded by a group I intron [15] with relatively short recognition sites, was chosen to modify the selection marker. We first generated a selection marker containing plasmid with only one I-PpoI site next to the selection marker (Appendix A). It worked almost in all our cloning setups, meaning that at least one or two positive clones can be identified, with some cases even 100% positive rate were observed. We were satisfied with this construct, but still curious about the effect of the selection marker that can be totally released from the adenoviral genome. Therefore, we generated a second plasmid with an I-PpoI flanked selection marker (Appendix A). Indeed, no significant difference was observed between the two versions.

We chose I-PpoI as the first candidate enzyme to accomplish this strategy based on flanking the selection marker with restriction enzyme recognition sites. However, there are certain limitations, which need to be considered. In the following work practice, we encountered adenovirus genomes already containing the I-PpoI site in previously inserted DNA sequences. To surmount this limitation, we decided to generate a third counter-selection marker plasmid with an AbsI site flanking (Appendix A). AbsI is a non-cutter in all six species except species D (Table 1). From our observation, the AbsI site flanking construct worked similarly as the two previous versions. Therefore, we strongly suggest individual users to carefully check their aimed sequence before they choose the enzyme-containing selection marker. In the case that neither I-PpoI nor AbsI are suitable for cloning, one can either construct an individual selection marker cassette following the method described in the Appendix A, or simply add-in an appropriate enzyme cutting site during primer design. 

LCHR is mediated by transient expression of the lambda phage recombinase (Redα and Redβ) from Red operon, to promote homologous recombination between a linear and circular DNA molecule in *E. coli* [37]. Most recombination protocols rely on Red proteins from lambda phage to mediate recombination between a circular and linear sequence, but poor efficiency for sequences with larger size (>10 kb) limits throughput for many applications. In contrast, LLHR is promoted by the full-length Rac prophage protein RecE and its partner RecT with high efficiency [38]. For both strategy and process, the linear DNAs are introduced by electroporation and serve as the substrates to introduce genetic change next to the region of homologous recombination. However, LCHR and LLHR are mechanistically different: LCHR recombination mediated by the Red proteins occurs at the replication fork and therefore requires ongoing replication, while replication is not required for RecET mediated LLHR, because it is started by simple annealing between two single-stranded regions [38,39,40]. In the current protocol, by introducing the universal non-cutter I-PpoI to the counter-selection marker, we convert the less efficient LCHR to highly efficient LLHR. In contrast to the LLHR method with endogenously expressed RecE and RecT, the NEB Gibson Assembly^®^ and NEBuilder^®^ HiFi DNA share a similar principle based on in vitro “homologous recombination” or assembly. The reaction is carried out under isothermal conditions using three enzymatic activities: a 5′ exonuclease generates long overhangs, a polymerase fills in the gaps of the annealed single strand regions, and a DNA ligase seals the nicks of the annealed and filled-in gaps [41]. In addition, NEBuilder has 3′ and 5′ -end mismatch remove function making it capable of up to 10 base pairs mismatches [42]. It is of note that the Gibson Assembly method has been widely adopted in synthetic biology projects including adenovirus engineering. For example, fowl adenovirus 4 genome was cloned into a pBR322 plasmid backbone using Gibson Assembly [34]. Moreover, site-directed modification of an adenoviral vector was also achieved by combination of Gibson Assembly and restriction-ligation [43,44]. However, compared to LLHR that is suitable for any targeting region for seamless modification, adenoviral vector engineering utilizing assembly methods is much more limited to locations that have manageable restriction enzyme sites in the near region.

It is of note, there are several factors that could influence the success rate of LLHR. First, transformation efficiencies of electroporation need to be mentioned. To this end, high quality DNA and *E.coli* preparations are essential [45]. Secondly, we observed that the L-arabinose activity has a strong influence on the recombination success rates. Although L-arabinose is recognized as being relatively stable [46], if unsatisfied cloning efficiencies are observed especially during selection marker insertion step, we strongly recommend to check and/or renew the L-arabinose stock solution. Finally, the cloning design represents a decisive factor for LLCR. For example, the sequence chosen as homology arm (HA) needs to be unique to avoid unwanted recombination events. Regarding the length of the HA, we found that HA of around 50 bp in length gave reliable efficiencies associated with stability in production of oligonucleotides and PCR results. However, please also note that we and others have also previously shown that longer HA resulted in higher cloning efficiency [4,38]. Regarding the difference in cloning efficiency shown in Figure 6, we speculate that this observation is a result of the difference in insert length and complexity, as well as the accessibility of the aimed target position to be modified. It was our purpose to not only show cloning procedures with 100% success rates, but also some cases with lower numbers of positive clones, because this was what we observed in our routine laboratory practice.

Taken together, the HEHR strategy and relative novel counter-selection marker containing plasmids presented in this protocol will facilitate the adenovirus genome engineering, therefore boosting the broad applications using adenovirus vector as vaccine to combat infectious diseases, as oncolytic virus to treat cancer or gene delivery vector for gene therapy.

## Figures and Tables

**Figure 1 genes-13-02129-f001:**
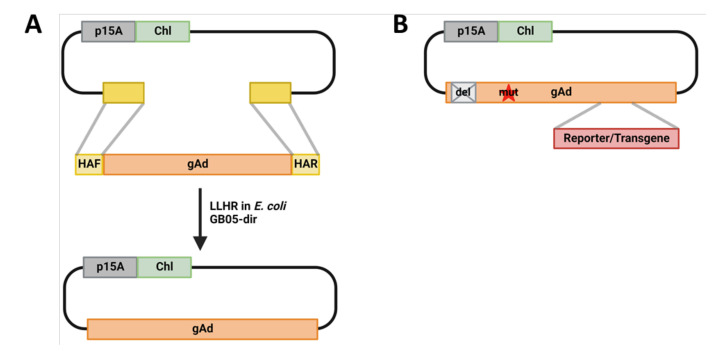
Adenoviral genome cloning and engineering. (**A**). Adenoviral genome direct cloning via linear–linear homologous recombination (LLHR). Adenoviral genome and PCR generated linear plasmid backbone flanked by homologous arms corresponding to the adenoviral genome (gAd) are co-electroporated into the *Escherichia coli* strain GB05-dir, in which the LLHR generates the adenoviral genome containing plasmid. (**B**). Whole-genome-wide seamless adenovirus engineering exemplified by introduction of mutations, deletions and reporter/transgene insertions. p15A, p15A plasmid origin of replication [13]; *Chl, chloramphenicol acetyltransferase*, Chloramphenicol resistance gene; gAd, adenovirus genome; del, deletion; mut, mutation. Figures are created with BioRender.com.

**Figure 2 genes-13-02129-f002:**
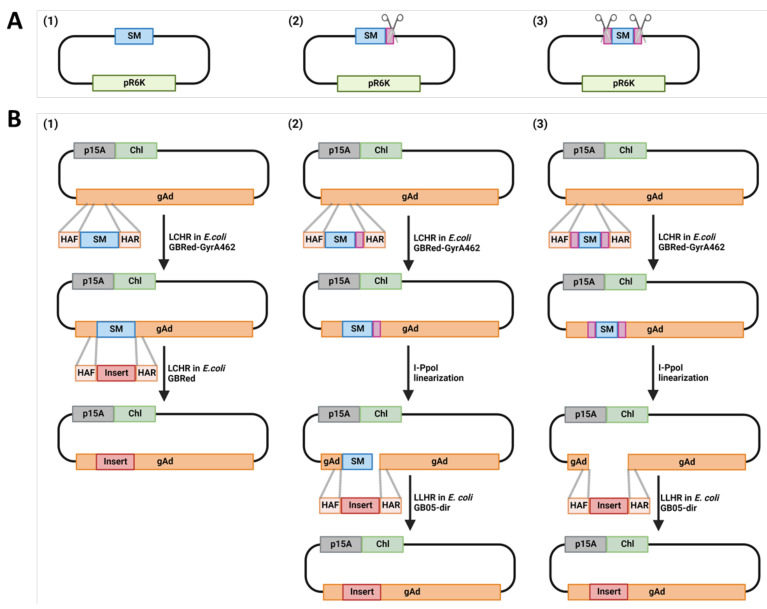
Homologous recombination strategy for adenovirus genome engineering. (**A**). The three types of selection marker (SM)-containing PCR templates for counter-selection are displayed. (1) SM only; (2) SM with a single flanking I-PpoI site; (3) SM with I-PpoI flanking both ends of the SM, for detail, see also Appendix A. (**B**). The three counter-selection strategies with different SM PCR-templates. All strategies shared the same first step: ampicillin-mediated positive selection to insert the selection marker cassette (*ccdB-Amp*) generated from different PCR-templates as shown in A. The small pink rectangles next to SM in B (2) and (3) show the I-PpoI sites. The adenoviral genome containing plasmid and the *ccdB-Amp* cassette flanked by homologous arms to the target region of the adenoviral genome are co-electroporated into the *E. coli* strain GBRed-GyrA462, in which the λ Red–mediated linear–circular homologous recombination (LCHR) takes place to insert the *ccdB-Amp* cassette into the desired site. The three strategies differ on the counter-selection step: (1) counter-selection takes place in *E. coli* strain GBRed, the adenoviral genome containing plasmid with previously inserted *ccdB-Amp* cassette, and the insert flanked by homologous arm to target region of adenoviral genome are co-electroporated. Then, the λ Red-mediated linear-circular homologous recombination (LCHR) replaces the *ccdB-Amp* cassette with insert. In contrast to strategy (1), the counter-selection marker containing adenovirus plasmids are pre-cut with I-PpoI in the strategies (2) and (3), which expose the homologous arm on one (2) or both sites (3). Then, the linearized adenovirus genome containing plasmid, and the insert flanked by homologous arms to the target region of the adenoviral genome are co-electroporated into the *E. coli* strain GB05-dir, in which the RecET recombinase is induced with L-arabinose to express, enabling linear-linear homologous recombination (LLHR) to replaces the *ccdB-Amp* cassette with aimed insert. p15A, p15A plasmid origin of replication [13]; chl, chloramphenicol acetyltransferase, chloramphenicol resistant gene; SM, *ccdB- ampicillin* double selection marker; pR6K, plasmid origin of replication that requires *pi* gene for replication [17,18]; HAF/R, homologous arm forward/reverse; gAd, adenovirus genome. Figures are created with BioRender.com.

**Figure 3 genes-13-02129-f003:**
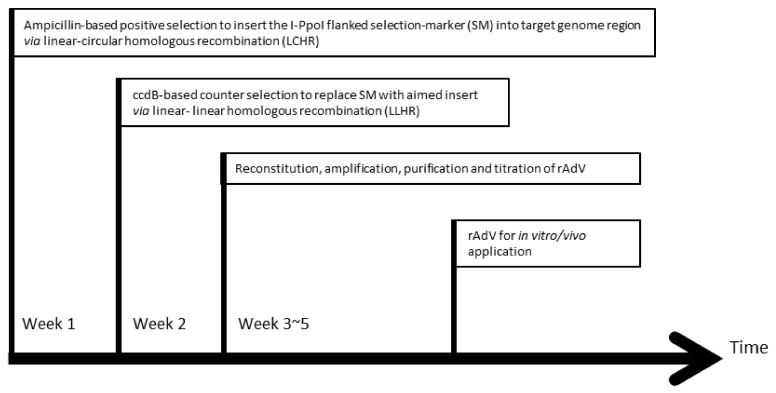
Timeline for adenovirus genome engineering with HEHR. It takes two weeks to generate a recombinant adenovirus vector. In the first week, a selection marker (SM) can be inserted into the target region of the adenoviral genome via linear-circular homologous recombination (LCHR); in the second week, the selection marker in the target site is replaced by the insert of choice via linear-linear homologous recombination (LLHR). Afterwards, the recombinant adenoviral vector can be reconstituted by transfection of the linearized plasmid into the producer cell line (such as HEK 293 cells) for further applications.

**Figure 4 genes-13-02129-f004:**
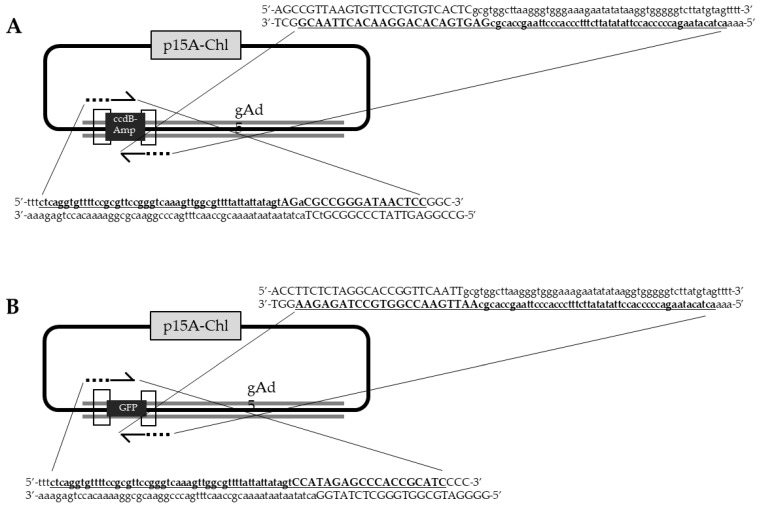
Example of oligonucleotide design for adenovirus genome engineering: transgene insertion into E1 region of adenovirus type 5 (Ad5). (**A**) Counter-selection marker (SM) *ccdB-Amp* insertion. (**B**) Transgene, here GFP, insertion. Forward- and reverse-primers are bold and underlined, capital letters are primer binding sequences, while lower case are homology arms.

**Figure 5 genes-13-02129-f005:**
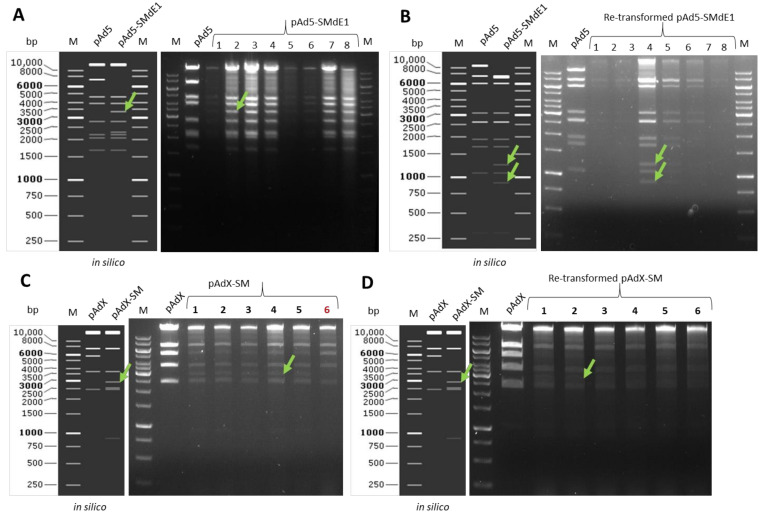
Example of plasmid restriction enzyme digest after positive-selection. Indicated with green arrows are the bands appearing only in positive clones. (**A**). The plasmid digest pattern is exemplified by selection marker (SM) insertion into E1 region of Ad5. The digest of both parental plasmid pAd5 and the aimed plasmid pAd5-SMdE1 with PvuI. All plasmids are positive for SM insertion evidenced from appearance of the 3.4 kb band. These bands with size range 5–10 kb could be contamination of *E.coli* genome. Therefore, the expected disappearance of ~7 kb band was not assessable. (**B**). The digest of both parental plasmid pAd5 and the aimed plasmid pAd5-SMdE1 after re-transformation, here the restriction enzyme DraIII was used, all clones were identified as clean-single-type correct plasmid. (**C**). The digest of both parental plasmid pAdX and the aimed plasmid pAdX-SM, here the restriction enzyme ApaLI was used. Five from six checked plasmids isolation were positive, evidenced from appearance of the 3 kb band. The appearance of the ~1 kb band was detectable in earlier gel-running time. Indicated in red is the digestion pattern of an incorrect clone. (**D**). The digest of both parental plasmid pAdX and the aimed plasmid pAdX-SM after re-transformation, here the restriction enzyme ApaLI was used, all clones were identified as clean-single-type correct plasmid.

**Figure 6 genes-13-02129-f006:**
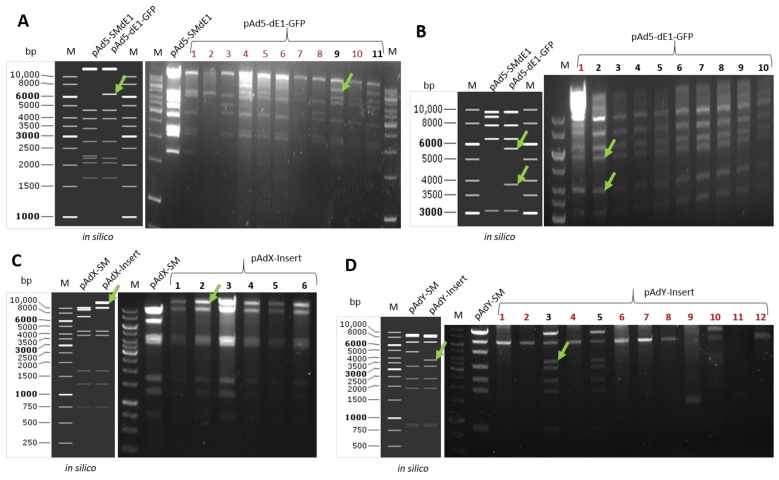
Example of plasmid restriction digest after counter-selection. Indicated with green arrows are the bands appearing only in positive clones. The plasmid digest pattern is exemplified by transgene (GFP) insertion into E1 region of Ad5. Indicated with red numbers are the clones with a wrong digestion pattern. (**A**). The digest of both parental plasmid pAd5-SMdE1 and the aimed plasmid pAd5-dE1-GFP with PvuI. Plasmids number 9 and 11 are positive for transgene insertion evidenced by the additional 6.1 kb band and the absence of the 3.4 kb band. Then, the two plasmids were re-transformed into normal competent bacteria. (**B**). The digest of both parental plasmid pAd5-SMdE1 and the aimed plasmid pAd5-dE1-GFP after re-transformation, here the restriction enzyme DraIII was used, all clones but not the clone number 1 were identified as clean-single-type correct plasmid. (**C**). The digest of both parental plasmid pAdX-SM and the aimed plasmid pAdX-Insert with BspHI, all six checked clones were positive evidenced by the bands shift. (**D**). The digest of both parental plasmid pAdY-SM and the aimed plasmid pAdY-Insert with SmaI, two (number 3 and 5) out of 12 checked clones were positive evidenced from the right digest bands pattern.

**Table 1 genes-13-02129-t001:** Non-cutters scanning of human adenoviral genome.

HAdV Species	AbsI	AhlI	AsiSI	BcuI	CpoI	CspI	FseI	I-CeuI	I-PpoI	I-SceI	MauBI	MssI	PacI	PI-PspI	PI-SceI	PmeI
A	X				X	X		X	X	X				X	X	
B	X		X					X	X	X		X		X	X	X
C	X							X	X	X				X	X	
D								X	X	X				X	X	
E	X	X	X	X				X	X	X			X	X	X	
F	X							X	X	X	X			X	X	
G	X						X	X	X	X			X	X	X	
**Shared**								X	X	X				X	X	
**HAdV Species**	**RgaI**	**RigI**	**RsrII**	**Rsr2I**	**SfaAI**	**SbfI**	**SdaI**	**SgfI**	**SgrDI**	**SmiI**	**SpeI**	**SrfI**	**Sse8387I**	**SwaI**	**total**	
A			X	X											10	
B	X				X			X		X		X		X	15	
C									X						7	
D	X				X			X							8	
E	X				X			X		X	X	X		X	17	
F						X	X						X		10	
G		X				X	X		X	X			X	X	15	
**Shared**															5	

The Non-cutters scanning was performed with SnapGene (Software number: SnapGene^®^ 2.5; Creator: Dotmatics, Boston, Massachusetts 02114, USA, https://www.snapgene.com/, accessed on 9 November 2022).

**Table 2 genes-13-02129-t002:** Reagents required for efficient adenovirus genome engineering.

Reagents	Supplier	Specific Handling	Storage Conditions
AbsI	GeneOn		−20 °C
Agarose	Peqlab		RT
Ampicillin sodium salt	Carl Roth	Stock solution in 70% ethanol (−20 °C)	4 °C
Ampuwa	Fresenius Kabi		RT
Chloramphenicol	Carl Roth	Stock solution in 70% ethanol (−20 °C)	RT
DNA ladder 1 kb	Peqlab		−20 °C
*E. coli* GB05-dir	Gene Bridge	Streak on LB-plate for use	−80 °C
*E. coli* GBRed-GyrA462	Gene Bridge	Streak on LB-plate for use	−80 °C
*E. coli* GBdir-pir116-gyrA462	Gene Bridge	Streak on LB-plate for use	−80 °C
Electroporation cuvettes 1 mm (reusable)	Cell projects		RT
Ethanol	Carl Roth		RT
Ethidium bromide	Carl Roth	Toxic, handing with care	4 °C
Gel and PCR cleanup	BioBudget		RT
Glycogen	Carl Roth	Stock solution in ddH_2_O (−20 °C)	4 °C
I-PpoI	Promega		−20 °C
Isopropanol	Carl Roth	Toxic, handing with care	RT
L-arabinose	Sigma-Aldrich	10% stock solution (−20 °C)	RT
Lysogene Broth (LB) powder	Carl Roth		RT
Nitrocellulose filters (0.025 µm)	Merck-Millipore		RT
Petri dish 10 cm	Sarstedt		
Phase Lock Gel™ heavy	VWR		RT
Phenol:chloroform:isoamyl-alcohol (25:24:1)	Carl Roth	Work in fume hood, dispose of all waste as hazardous waste	4 °C
PrimeSTAR^®^ Max DNA Polymerase	TAKARA		−20 °C
pR6K-ccdB-Amp	[7]		−20 °C
pR6K-ccdB-Amp-(IP)	This study		−20 °C
pR6K-(IP)-ccdB-Amp-(IP)	This study		−20 °C
pR6K-(AB)-ccdB-Amp-(AB)	This study		−20 °C
pR6K-hyg-spect-PBs	[19]		−20 °C
Primer (≤50 nucleic acids)	Eurofins Genomics		−20 °C
Primer (>50 nucleic acids)	Sigma-Aldrich	With HPLC purity	−20 °C
Restriction enzymes	NEB		−20 °C
Sodium acetate	Carl Roth		RT
Streptomycin	Carl Roth		RT

RT, room temperature.

**Table 3 genes-13-02129-t003:** Composition of PrimeSTAR DNA polymerase PCR reaction mixture.

Total	Amount	Final Concentration
Template pR6K-(IP)-ccdB-Amp-(IP)	1 µL	~1 ng/µL
PrimeSTAR Max Premix (2x)	50 µL	1×
HR-primer-fwd	5 µL	0.5 µM
HR-primer-rev	5 µL	0.5 µM
deionized H_2_O	40 µL	

**Table 4 genes-13-02129-t004:** PrimeSTAR DNA polymerase PCR condition.

Parameters	Temperature	Time	Cycles
Denaturation	98 °C	10 s	35
Annealing	55 °C	5 s	
Elongation	72 °C	5 s/kb	

**Table 5 genes-13-02129-t005:** Example of I-PpoI digestion of pAd-SM plasmid.

	Content
DNA (µg)	~3–5
dH2O (µL)	*
I-PpoI (µL)	1
Acetylated BSA 100× (µL)	1.5
I-PpoI-buffer (µL)	15
Total (µL)	150

*:The amount of the water will depend on the DNA concentration, so adjust the water volume to complete the total volume after adding of all materials to equal 150 µL.

## Data Availability

All used *E. coli* strains and plasmids generated/used in this manuscript are available upon request.

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
