# Peer review of "HEHR: Homing Endonuclease-Mediated Homologous Recombination for Efficient Adenovirus Genome Engineering"

_genes, 2022, doi:10.3390/genes13112129_

Round 1

Reviewer 1 Report

Authors proposed a new method to engineer Ad vector with genome modifications more efficiently by using homing endonuclease.  This concept is not too complicated, and it is able to incorporate any modifications or genes in any places of Ad gene. However, there are some unclear points.

1: There are 5 of endonuclease which were filtered out as the non-cutters, and authors chose I-PpoI. Is there any specific reason to picked I-Ppol? I think it’s better to have more detail explanation of why I-Ppol is better than others.

2:  For the homologous recombination efficiency, is there any difference between a single flanking I-PpoI site vs 2 I-PpoI flanking in both ends of the SM? If there is difference, the data should be in the paper.

3: Authors generated a third counter-selection marker plasmid with AbsI site flanking. Is this vector working the same as I-Ppol vector? How was the success rate of homologous recombination? Do you have any data of this vector?

 4: In the abstract section, Authors said that “We found that the I-PpoI pre-treatment of counter-selection containing parental plasmid increased the homologous recombination efficiency up to 100 %.” In Figure 6, AdX-based vector showed 100% positive, but Ad5-based or AdY-based vector showed low positive rate, 2/11 or 2/12 respectively.  Is there any reason why AdX-based one has high positivity?

 5: Recently, there are several papers that are using Gibson cloning to engineer the adenovirus vectors. Compared with this method, what is the merit of your LLHR method?

Author Response

Response to reviewer 1:

We are very grateful to this reviewer for the positive statement and constructive suggestions, which helped to significantly improve the manuscript.

Please find the answers to the specific points below.

1: There are 5 of endonuclease which were filtered out as the non-cutters, and authors chose I-PpoI. Is there any specific reason to picked I-Ppol? I think it’s better to have more detail explanation of why I-Ppol is better than others.

Response: we agree that the description regarding the choice of I-PpoI was not sufficient. As suggested and to address this comment, we added the following paragraph to the revised manuscript:

 “We chose this I-PpoI based on two simple reasons: This endonuclease has the shortest recognition sequence (15 bp) among the five available enzymes, which makes the selection marker plasmid construction convenient. Moreover, the other four endonucleases were quite often used as enzymes to release the adenoviral genome for virus rescue, such as I-CeuI and I-SceI already positioned in our p15A-based vector backbone”

2: For the homologous recombination efficiency, is there any difference between a single flanking I-PpoI site vs 2 I-PpoI flanking in both ends of the SM? If there is difference, the data should be in the paper.

Response: thank you for this point; to address this issue we added the following paragraph in the discussion section:

We first generated a selection marker containing plasmid with only one I-PpoI site next to the selection marker (Supplementary Methods, Figure S1A). It worked almost in all our cloning setups, meaning that at least one or two positive clones can be identified, with some cases 100% positive rate were observed. We were satisfied with this construct, however, still curious about the effect of the selection marker that can be totally released from the adenoviral genome. Therefore, we generated a second plasmid with an I-PpoI flanked selection marker (Supplementary Methods, Figure S1B). Indeed, no significant difference was observed between the two versions.”

In our current labor practice, some people prefer to use the single flanking I-PpoI site, while others prefer the two I-PpoI flanking version.

3: Authors generated a third counter-selection marker plasmid with AbsI site flanking. Is this vector working the same as I-Ppol vector? How was the success rate of homologous recombination? Do you have any data of this vector?

Response: similar success rates for homologous recombination were observed from the AbsI site flanking construct and the two previous versions based on I-PpoI. To address this comment we added the following paragraph in the revised discussion section:

From our observation, the AbsI site flanking construct worked similarly as the two previous versions. Therefore, we strongly suggest individual users to check carefully their aimed sequence before they choose the enzyme-containing selection marker. In the case that neither I-PpoI nor AbsI are suitable for cloning, one can either construct an individual selection marker cassette following the method described in the Supplementary Methods, or simply add-in an appropriate enzyme cutting site during primer design.”

4: In the abstract section, Authors said that “We found that the I-PpoI pre-treatment of counter-selection containing parental plasmid increased the homologous recombination efficiency up to 100 %.” In Figure 6, AdX-based vector showed 100% positive, but Ad5-based or AdY-based vector showed low positive rate, 2/11 or 2/12 respectively.  Is there any reason why AdX-based one has high positivity?

Response: there is no special reason that the AdX-based vector has higher positivity. We prefer to show the readers not only the 100% positive case, but also some cases with only one or two positive clones, because this was what we observed in our laboratory practice.

Meanwhile, the reviewer pointed out a very interesting and critical question regarding homologous recombination efficiency. We now discuss the major factors that could influence the success rate of LLHR and therefore added the following paragraph to the discussion section:

“It is of note, there are several factors that could influence the success rate of LLHR. At first, transformation efficiencies of electroporation need to be mentioned. To this end, high quality DNA and E.coli preparations are essential [44]. Secondly, we observed that the L-arabinose activity has a strong influence on the recombination success rates. Although L-arabinose is recognized as being relatively stable [45], if unsatisfied cloning efficiencies are observed especially during selection marker insertion step, we strongly recommend to check and/or renew the L-arabinose stock solution. At last, the cloning design represents a decisive factor for LLCR. For example, the sequence chosen as homology arm (HA) needs to be unique to avoid unwanted recombination events. Regarding the length of the HA, we found that HA of around 50 bp in length gave reliable efficiencies associated with stability in production of oligonucleotides and PCR results. However, please also note that we and others have also previously shown that longer HA resulted in higher cloning efficiency [4,37]. Regarding the difference in cloning efficiency showed in Figure 6, we speculate that this observation is a result of the difference in insert length and complexity, as well as the accessibility of the aimed target position to be modifed.”

 5: Recently, there are several papers that are using Gibson cloning to engineer the adenovirus vectors. Compared with this method, what is the merit of your LLHR method?

Response: as the reviewer suggested, we further discussed the assembly methods and point out one advantage of LLHR compared to assembly methods. We also added the following paragraph in the revised discussion section:

In contrast to the LLHR method with endogenously expressed RecE and RecT, the NEB Gibson Assembly® and NEBuilder® HiFi DNA share a similar principle based on in vitro “homologous recombination” or assembly. The reaction is carried out under isothermal conditions using three enzymatic activities: a 5’ exonuclease generates long overhangs, a polymerase fills in the gaps of the annealed single strand regions, and a DNA ligase seals the nicks of the annealed and filled-in gaps [40]. In addition, NEBuilder has 3’ and 5’ -end mismatch remove function making it capable of up to 10 base pairs mismatches [41]. It is of note that the Gibson Assembly method has been widely adopted in synthetic biology projects including adenovirus engineering. For example, fowl adenovirus 4 genome was cloned into a pBR322 plasmid backbone using Gibson Assembly [33]. Moreover, site-directed modification of an adenoviral vector was also achieved by combination of Gibson Assembly and restriction-ligation [42,43]. However, compared to LLHR that is suitable for any targeting region for seamless modification, adenoviral vector engineering utilizing assembly methods is much more limited to locations that has manageable restriction enzyme sites in the near region.

Reviewer 2 Report

Manuscript by Schröder et al., describe a new trick how to get recombinant Ad vector with minimal pain and high efficiency. Everyone who has worked with recombinant Ad vectors knows the challenges and enormous time spent to get correct recombinant Ad vector in his/her hands. So, every small trick which can help to make these vectors faster is extremely welcomed.

The manuscript is very well written, has all the details in place and shows the primary raw data as well. One should give a credit to the authors by really going into small details of the whole procedure and really highlighting the pros and cons. Thank you for that! 

I just have a few minor comments what the authors can use to improve the manuscript.

1) I am not sure that it is important to include Table 1 in the main manuscript. The message about I-PpoI is there, but the same Table could be also in Supplementary. In this way the nice explanation in Fig. 2 will be more highlighted, now it stays in the shadow of table 1. Also it would be nice to include homepage of the used SnapGene program as for example  have never heard about it as I use CLC Bio etc.

2)Figure 5. I love the idea to include the in silico cleavage pattern beside the real cleavage pattern. However, it would be good to discriminate these two. Just label the in silico pattern somehow that it is in silico or theoretical patter on the figure. Also (it might be an overkill) it would be more elegant to point out with white arrows the bands which are "popping-up" in correct clones. It also remains unclear to me why in the Fig5 legend the authors claim that additional 6.1 kb band discriminates the correct clone from wrong one? Can one not use the disapperance of the 3.5 band as the diagnostic read-out?

3) Will the authors deposit their plasmid to Addgene? Would be nice if it is mentioned in the manuscript

Author Response

Comments to reviewer 2:

We are appreciating the specific suggestions raised by this reviewer. Please find the answers to these comments below.

Minor comments

1. The reviewer suggested shifting Table 1 from the main manuscript to the Supplementary Material section, because Figure 2 will then be more highlighted. However, we believe that this table represent important information, and decide to keep it in the major text to maintain its value. We hope that the reviewer can understand our choice.

2. As also recommended by the reviewer, we included the homepage of SnapGene, https://www.snapgene.com/ in the revised version of the manuscript.

3. The reviewer asked to label the in silico pattern in Figure 5 with arrows to discriminate it from the real cleavage pattern. This was implemented in the revised version of the main manuscript for Figures 5 and 6 where the “popping-up” bands are now indicated with green arrows. Furthermore, the reviewer asked if, besides the appearance of the 6.1 kb band in figure 6A, the disappearance of the 3.5 kb band is also an indicator for positive clones. Regarding this latter point the reviewer is correct and this additional information was added to the respective figure legend.

4. The reviewer asked whether the plasmid will be uploaded in Addgene and if we could mention that in the manuscript.

Uploading in Addgene is a good idea to share material. We agree with the reviewer, however, we also added the following “Material Availability Statementin the revised version of the manuscript:

“All used E. coli strains and plasmids generated / used in this manuscript are available upon request.”